# Clinical characteristics, comorbidities, and correlation with advanced lipedema stages: A retrospective study from a Swiss referral centre

Xhyljeta Luta[1]*, Giacomo Buso[1,2], Enrica Porceddu[1], Roxani Psychogyiou[1], Sanjiv Keller[1], Lucia Mazzolai[1]

1 Department of Angiology, Lausanne University Hospital (CHUV), Lausanne, Switzerland, 2 Division of Internal Medicine, Department of Clinical and Experimental Sciences, ASST Spedali Civili Brescia, University of Brescia, Brescia, Italy

* xhyljeta.luta@chuv.ch

## Abstract

### Introduction

Lipedema is a chronic condition involving abnormal fat deposition in the lower limbs, often underdiagnosed, and poorly understood. We examined the epidemiological and clinical characteristics of a large patient cohort in Switzerland and their associations with disease severity.

### Methods

We included women aged 18 and over with lipedema at Lausanne University Hospital (CHUV), Switzerland. Demographic and clinical data, including disease type, stage, symptoms, and comorbidities, were collected. Descriptive statistics were used to summarise the data, and logistic regression was employed for analysis.

### Results

A total of 381 females (mean age 41.9 years) were included, mostly classified as type III (48.3%) and IV (30.2%) lipedema. In our population, 26.6% of patients were classified as stage 1, 44.5% as stage 2, and 28.9% as stages 3–4. Family history was reported in 49.9%, with symptoms often starting during adolescence (62.2%). Pain affected 87.9%, and quality of life (QoL) was significantly reduced, with 71.5% reporting low physical and 67.4% low mental well-being. Comorbidities were present in 92.1%, increasing with advanced disease stage, with chronic venous disease (86.2%) and obesity (51.7%) being the most common. Univariate analysis showed advanced lipedema was associated with age (OR: 1.07, 95% CI: 1.05–1.09), BMI (OR: 1.24, 95% CI: 1.19–1.29), and comorbidities (OR: 1.59, 95% CI: 1.39–1.81). Multivariate analysis confirmed age (OR: 1.06, 95% CI: 1.04–1.08) and BMI (OR: 1.22, 95% CI: 1.17–1.28) as correlates with disease stage.

**Data availability statement:** The data are not publicly available due to privacy and ethical restrictions. Data are available upon request. Requests for data access can be directed to the following contact referencing our project ID: CER-VD, BASEC 2023-02221: Commission cantonale d'éthique de la recherche sur l'être humain Avenue de Chailly 23 1012 Lausanne Tel: + 41 21 316 18 36 E-Mail: scientifique. cer@vd.ch.

**Funding:** The author(s) received no specific funding for this work.

**Competing interests:** The authors have declared that no competing interests exist.

## Conclusions

Our study highlights frequent comorbidities in patients with lipedema, including chronic venous disease, obesity, and mental health conditions such as anxiety and depression. The distribution of comorbidities supports the need for tailored management. The correlation between disease stages, age, and BMI suggests potential progression, warranting confirmation through prospective studies.

## Introduction

Lipedema is a chronic condition primarily affecting female sex, characterised by abnormal subcutaneous fat accumulation, mainly in the lower limbs and occasionally the arms [1]. It is classified into five types based on regional fat distribution and three to four stages according to severity. The prevalence of the condition remains uncertain, ranging from 1:72,000 [2] in the general population to 18.8% among patients with lower limb enlargement at specialised clinics [3]. Although rare, cases in men have also been reported [4,5].

The causes of lipedema remain unclear. Genetic transmission predisposition is possible, as lipedema often runs in families [2]. Previous studies have reported a positive family history in 16% to 64% of affected patients [6]. However, subsequent research has not identified genes strongly linked to the disease. Hormonal factors [7], chronic inflammation [8], microcirculation disorders [9], and adipocyte dysregulation [10] may also contribute to the development and progression of lipedema, though evidence remains limited in this area.

The diagnosis of lipedema is almost exclusively based on clinical features, which can be subjective and not always consistently reliable. Furthermore, symptoms often overlap with other similarly presenting conditions, such as lymphoedema, obesity, or chronic venous insufficiency [11–13], leading to delays in diagnosis and inappropriate treatment in many cases [14].

The main goals of treatment are to alleviate symptoms such as heaviness, pain, and swelling in the affected limbs, improve patients' quality of life (QoL), and prevent disease progression. Treatment options include conservative approaches such as regular use of flat-knit elastic compression hosiery combined with decongestive therapy. For selected patients with persistent symptoms despite optimal conservative care, surgical interventions, such as liposuction, may be considered [15]). Although there is no international consensus on the management of lipedema, several guidelines exist across different countries. These guidelines universally emphasise the importance of a comprehensive approach that include lifestyle management [16–21].

Despite the growing scientific literature on lipedema in recent years, only a few studies have comprehensively evaluated the clinical characteristics of patients with lipedema [22–25], most of which were based on surveys rather than clinical assessments. Accurately characterising individuals with lipedema in a large population can significantly enhance our understanding of the condition, provide valuable insights for diagnosis, and support the development of tailored and effective treatment plans. Therefore, the aims of our study were: (i) to describe the epidemiological and clinical characteristics of a large cohort of patients with lipedema, assessed at a Swiss referral centre, covering the full spectrum of disease severity; and (ii) to explore the associations between these characteristics and the severity of the condition.

## Materials and methods

### Study design and setting

This is a retrospective study conducted at the Centre for Malformations and Rare Vascular Diseases (CMRVD) at Lausanne University Hospital (CHUV), Switzerland. The CMRVD is

a Swiss referral centre for the management of lipedema, with over 500 patients currently in regular follow-up.

## Study population

We included women aged 18 and over with lipedema who attended a routine consultation at our centre. Diagnosis and classification were conducted by specialists using pre-defined criteria [15,26,27]. Inclusion criteria were: (1) bilateral, symmetrical fat accumulation in the limbs with minimal or no involvement of hands and feet; (2) minimal improvement with weight loss strategies (excluding ketogenic diets); (3) pain, tenderness, bruising, sensitivity, or fatigue in affected limbs; (4) limited or no pitting oedema; and (5) no relief of pain or discomfort with limb elevation.

## Data sources and definitions

We analyzed electronic health record data collected from July 2017 up to December 2023, obtained from the RAVAD (Registry of rAre vascular disease) registry. Data were accessed in January 2024. Access to the project data was limited to authorized persons (XL, LM). Established in 2017 at the CMRVD (CHUV), the registry systematically includes patients with rare vascular diseases (suspected or confirmed), aiming to enhance understanding of disease etiology and optimize their management.

We collected data on patient demographics and anthropometrics, including age, sex, employment status, body weight (kg), height (cm), and Body Mass Index BMI (calculated as weight divided by height squared, classified according to the World Health Organization (WHO) criteria) [28], and lipedema characteristics (e.g., onset, clinical signs, type, stage). Information on cardiovascular risk factors (smoking, hypertension, diabetes, hypercholesterolaemia, alcohol consumption), relevant comorbidities, family history, and medical or surgical history was also gathered. WHO BMI categories were classified as follows: normal weight (18–24.9 kg/m$^2$), overweight (25–29.9 kg/m$^2$), obesity class I (30–34.9 kg/m$^2$), class II (35–39.9 kg/m$^2$), and class III (≥40 kg/m$^2$) [28]. Blood pressure categories were defined according to the 2017 ACC/AHA Hypertension Guidelines, classifying measurements as normal (<120/<80 mmHg), elevated (120–129/<80 mmHg), stage 1 hypertension (130–139/80–89 mmHg), and stage 2 hypertension (≥140/≥90 mmHg) [29].

Lipedema was classified into five types based on anatomical location: type I (fat distribution in the pelvis and buttocks); type II (fat accumulation from the pelvis to the knees); type III (fat distribution from the hip to the ankle with a typical "cuff sign" at the ankle, sparing the dorsal foot); type IV (fat distribution from the shoulders to the wrists); and type V (primarily affecting the calves). The disease was further classified into four stages based on subcutaneous fat and skin alterations: stage 1 (smooth skin over a thick, nodular hypodermis); stage 2 (uneven skin with palpable, pearl-sized nodules, often described as "orange peel skin"); stage 3 (lobular protrusions of skin, fat, and fascial tissue causing significant deformities, particularly around the thighs and knees); and stage 4 (associated with lymphoedema) [1]. The presence of lymphoedema was determined using current clinical and radiological criteria [30]. For this study, we defined advanced lipedema as disease stage 3-4.

Comorbidities were extracted from medical records and grouped into ten categories: cardiac, pulmonary, gastrointestinal, hepatic, rheumatic, neurological, psychiatric, vascular, ophthalmic, and others. Vascular conditions were further classified into eight diagnostic categories: atherosclerotic peripheral artery disease, venous thromboembolism, connective tissue disorders, cerebrovascular disease, coronary artery disease, chronic venous diseases, and other conditions. Joint hypermobility was defined as a Beighton score ≥ 5 [31].

We also collected QoL data using standardised questionnaires for 239 patients, including the SF-36 [32]) for general health, the Fatigue Severity Scale (FSS) [33]) for fatigue impact, the

Brief Pain Inventory (BPI) [34] for pain intensity, and the Hospital Anxiety and Depression Scale (HADS) [35] for anxiety and depression levels.

## Statistical analysis

We performed descriptive statistics to summarise the patients' sociodemographic and clinical characteristics. Continuous variables are presented as means with standard deviations (SD) and categorical variables are expressed as counts and percentages. We examined the distribution of patients' characteristics both in the entire cohort and according to disease stages. Comparisons across lipedema stages were conducted using chi-square tests (Pearson) for categorical variables (e.g., BMI classification, lipedema type, and comorbidities). Simple and multivariable linear regression analyses were performed to identify potential correlates of disease severity, including variables such as age, comorbidities, and BMI. Interaction effects between age and BMI were explicitly tested in this model to investigate whether the relationship between BMI and disease severity varied by age. Statistical significance was set at $p < 0.05$. Data management and analysis were conducted using Stata version 18 (StataCorp, College Station, TX, USA).

## Ethics approval

The study has been carried out in accordance with the ethical standards of the 1964 Declaration of Helsinki and its later amendments. This study was approved by the local ethics committee (CER VD, BASEC 2023-02221). All study participants provided their written informed consent for the use of their clinical data for research purposes.

## Results

Table 1 summarises patient demographics and clinical characteristics both overall and stratified by disease stage. The cohort included 381 female patients, with 101 (26.6%) in stage 1, 169 (44.5%) in stage 2, and 110 (28.9%) in stages 3-4 lipedema. Nearly half (48.3%) had type III lipedema, while 115 (30.2%) presented with upper limb involvement. The mean age was 41.9 years (SD: 12.5), and the mean BMI was 30.6 kg/m². According to WHO criteria, 194 patients (51.0%) were classified as having obesity class I or higher.

Mean age was highest in the stage 3-4 group (48.7 years p < 0.001) compared both to stages 1 (34.4 years) and 2 (41.9 years,). BMI increased significantly across stages (Fig 1), with mean BMI reaching 36.3 kg/m² in stage 3-4, compared to 26.4 kg/m² in stage 1 (p < 0.001) and 29.3 kg/m² in stage 2 (p < 0.001).

Normal blood pressure was observed in 31.2% of participants, while 42% had elevated levels. Stage 1 and stage 2 blood pressure were observed in 26% and 16.7% of participants with stage III-IV lipedema, respectively (p < 0.001). Over 60% of patients reported lipedema onset during adolescence, with 24.2% developing it later in life. Nearly half (49.9%) had a positive family history. The most frequently affected family member was the mother (53.2%), followed by grandmothers (30.0%), sisters and aunts (13.7% each), daughters (13.2%), and cousins (5.3%) (Fig 2).

Overall, 14.2% of participants reported high pain levels, 56.1% experienced significant fatigue, 64.4% reported anxiety, and 23.4% showed signs of depression. QoL assessments revealed low physical scores (PCS) in 71.5% and low mental scores (MCS) in 67.4%. There were no significant differences in lipedema onset, family history, or QoL measures across disease stages.

Table 2 summarises symptom prevalence across the entire study population, while Supplementary material (S1 Table) provides a breakdown of the data by disease stage. The most common symptoms were limb pain (87.9%), disproportionate limb enlargement (82.2%), easy bruising (77.7%), and limb heaviness (76%).

**Table 1. Baseline characteristics of the study population, overall and stratified by lipedema stage.**

| | Overall | Stage 1 | Stage 2 | Stage 3-4 | p-value[a] |
|---|---|---|---|---|---|
| | | (N = 101) | (N = 169) | (N = 110) | |
| **Age mean (SD)[b], y** | 41.9 (12.5) | 34.4 (10.1) | 41.9 (11.6) | 48.7 (12.0) | <0.001 |
| **BMI, mean (SD)** | 30.6 (8.3) | 26.4 (4.4) | 29.3 (4.8) | 36.3 (6.9) | <0.001 |
| Obesity class I | 107 (28.1%) | 20 (19.8%) | 53 (31.4%) | 34 (30.9%) | <0.001 |
| Obesity class II | 49 (12.9%) | 4 (4.0%) | 21 (12.4%) | 24 (21.8%) | |
| Obesity class III | 38 (10.0%) | 1 (1.0%) | 3 (1.8%) | 34 (30.9%) | |
| **Blood pressure** | | | | | |
| **SBP [c], mean (SD)** | 126.8 (14.8) | 124.6 (12.3) | 125.3 (13.8) | 131.4 (17.3) | 0.002 |
| **DBP [d], mean (SD)** | 80.2 (10.5) | 78.6 (11.4) | 78.48 (8.7) | 84.4 (11.01) | <0.001 |
| Normal | 104 (31.2%) | 32 (36.0%) | 55 (37.4%) | 17 (17.7%) | <0.001 |
| Elevated | 140 (42.0%) | 39 (43.8%) | 62 (42.2%) | 38 (39.6%) | |
| Stage 1 | 66 (19.8%) | 15 (16.9%) | 26 (17.7%) | 25 (26.0%) | |
| Stage 2 | 23 (6.9%) | 3 (3.4%) | 4 (2.7%) | 16 (16.7%) | |
| **Lipedema type** | | | | | |
| Type I | 9 (2.4%) | 6 (5.9%) | 2 (1.2%) | 1 (0.9%) | <0.001 |
| Type II | 47 (12.3%) | 19 (18.8%) | 23 (13.6%) | 5 (4.5%) | |
| Type III | 184 (48.3%) | 42 (41.6%) | 95 (56.2%) | 47 (42.7%) | |
| Type IV | 115 (30.2%) | 22 (21.8%) | 37 (21.9%) | 55 (50.0%) | |
| Type V | 26 (6.8%) | 12 (11.9%) | 12 (7.1%) | 2 (1.8%) | |
| **Lipedema stage** | | | | | |
| Stage 1 | 101 (26.6%) | -- | -- | -- | |
| Stage 2 | 169 (44.5%) | -- | -- | -- | |
| Stage 3-4 | 110 (28.9%) | -- | -- | -- | |
| **Family history** | 190 (49.9%) | 50 (49.5%) | 78 (46.2%) | 61 (55.5%) | 0.315 |
| **Disease onset** | | | | | |
| Childhood | 20 (5.2%) | 6 (5.9%) | 8 (4.7%) | 6 (5.5%) | 0.750 |
| Adolescence | 237 (62.2%) | 68 (67.3%) | 105 (62.1%) | 64 (58.2%) | |
| Adulthood | 46 (12.1%) | 10 (9.9%) | 21 (12.4%) | 15 (13.6%) | |
| Pregnancy | 38 (10.0%) | 8 (7.9%) | 18 (10.7%) | 12 (10.9%) | |
| Menopause | 8 (2.1%) | 1 (1.0%) | 2 (1.2%) | 5 (4.5%) | |
| Not specified | 32 (8.4%) | 8 (7.9%) | 15 (8.9%) | 8 (7.3%) | |
| **Comorbidities** | | | | | |
| 0 | 30 (7.9%) | 12 (11.9%) | 16 (9.5%) | 1 (0.9%) | <0.001 |
| 1-3 | 249 (65.4%) | 76 (75.2%) | 116 (68.6%) | 57 (51.8%) | |
| 4+ | 102 (26.8%) | 13 (12.9%) | 37 (21.9%) | 52 (47.3%) | |
| **Pain (n = 239)** | | | | | |
| Low (< 7) | 205 (85.8%) | 64 (97.0%) | 83 (82.2%) | 58 (80.6%) | 0.009 |
| High (>= 7) | 134 (14.2%) | 2 (3.0%) | 18 (17.8%) | 14 (19.4%) | |
| **Fatigue (n = 239)** | | | | | |
| Low (< 4) | 105 (43.9%) | 32 (48.5%) | 41 (40.6%) | 32 (44.4%) | 0.601 |
| High (>= 4) | 134 (56.1%) | 34 (51.5%) | 60 (59.4%) | 40 (55.6%) | |
| **Anxiety (n = 239)** | | | | | |
| No (<= 7) | 85 (35.6%) | 21 (31.8%) | 36 (35.6%) | 28 (38.9%) | 0.687 |
| Yes (>= 8) | 154 (64.4%) | 45 (68.2%) | 65 (64.4%) | 44 (61.1%) | |
| **Depression(n = 239)** | | | | | |
| No (<= 7) | 183 (76.6%) | 53 (80.3%) | 76 (75.2%) | 54 (75.0%) | 0.701 |
| Yes (>= 8) | 56 (23.4%) | 13 (19.7%) | 25 (24.8%) | 18 (25.0%) | |

*(Continued)*

**Table 1.** (Continued)

|  | Overall | Stage 1 | Stage 2 | Stage 3-4 | p-value[a] |
| --- | --- | --- | --- | --- | --- |
|  |  | (N = 101) | (N = 169) | (N = 110) |  |
| **PCS (SF-36) (n = 239)** [e] |  |  |  |  |  |
| Low QoL | 171 (71.5%) | 45 (68.2%) | 73 (72.3%) | 53 (73.6%) | 0.821 |
| Moderate QoL | 37 (15.5%) | 10 (15.2%) | 15 (14.9%) | 12 (16.7%) |  |
| High QoL | 31 (13.0%) | 11 (16.7%) | 13 (12.9%) | 7 (9.7%) |  |
| **MCS (SF-36)** [f] |  |  |  |  |  |
| Low QoL | 161 (67.4%) | 48 (72.7%) | 70 (69.3%) | 43 (59.7%) | 0.513 |
| Moderate QoL | 40 (16.7%) | 9 (13.6%) | 15 (14.9%) | 16 (22.2%) |  |
| High QoL | 38 (15.9%) | 9 (13.6%) | 16 (15.8%) | 13 (18.1%) |  |

[a]) P-values indicate differences between lipedema stages;

[b]) Standard Deviation;

[c]) Systolic Blood Pressure;

[d]) Diastolic Blood Pressure;

[e]) Physical Component Summary;

[f]) Mental Component Summary.

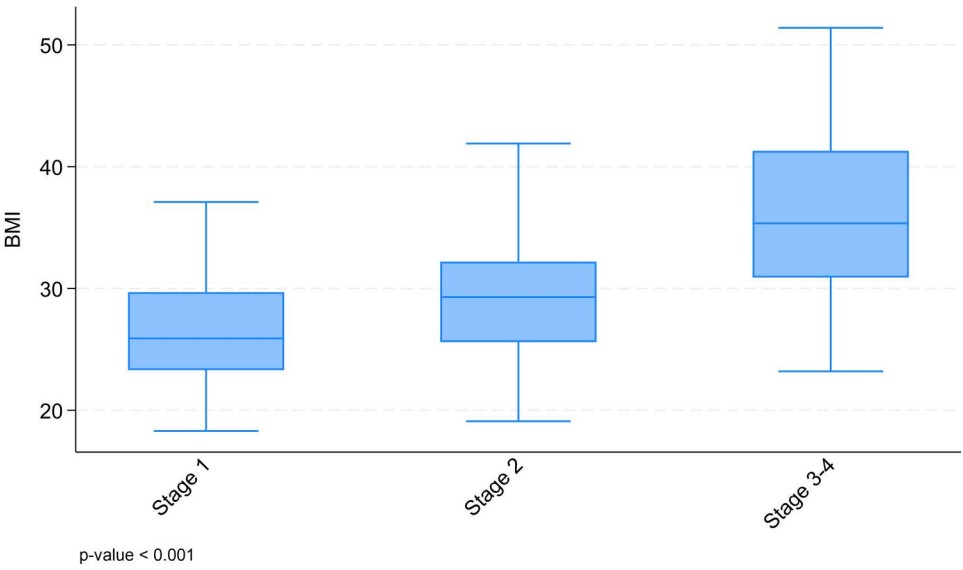

**Fig 1. BMI across lipedema stages: box plots with the median and interquartile range.**

Approximately 4% of patients had a positive Stemmer sign, while 4.2% had a Beighton score of 5 or higher, indicative of joint hypermobility. Symptoms did not differ significantly across disease stages (S1 Table).

## Distribution of comorbidities

Among participants, 7.9% reported no comorbidities, 65.4% had 1–3, and 26.8% had 4 or more, indicating that most experienced at least one additional health condition (Table 1). The number of comorbidities increased with advanced lipedema stages: 12.9% of those with stage I had 4 or more comorbidities, compared to 21.9% with stage II, and nearly half (47.3%) of those with advanced disease (p < 0.001).

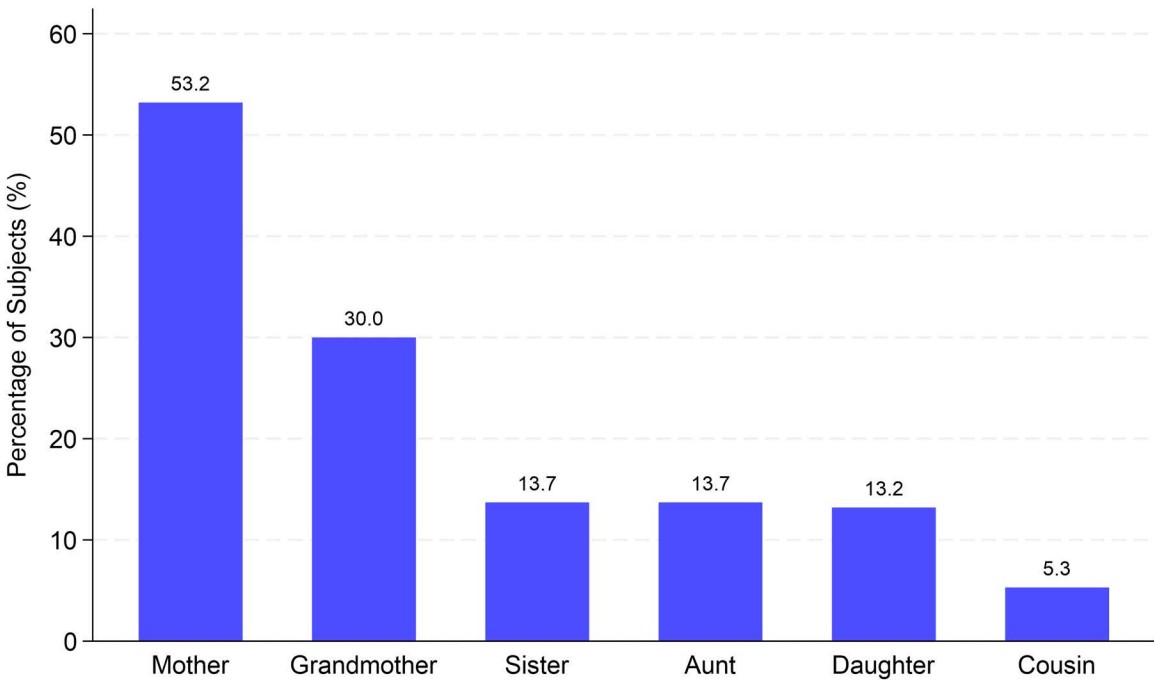

**Fig 2. Family history of lipedema by relative type.**

**Table 2. Prevalence of most common lipedema symptoms at baseline.**

| Symptoms | Summary |
|---|---|
| | (N = 381) |
| Easy bruising | 296 (77.7%) |
| Pain | 335 (87.9%) |
| Heaviness | 289 (75.9%) |
| Disproportionate enlargement of limbs | 313 (82.2%) |
| Stemmer positive | 15 (3.9%) |
| Hypermobility ( ≥ 5) [a] | 16 (4.2%) |

[a]) Hypermobility was classified into two groups: non-hypermobile (score < 5) and hypermobile (score ≥ 5).

Fig 3 shows the distribution of the most frequently reported comorbidities. Chronic venous disease was the most common, affecting 86.2% of the sample, followed by obesity at 51.7%. Other notable conditions included hypothyroidism and psychiatric disorders, each affecting 11.8% of patients. Less common conditions included rheumatic diseases (7.3%), hypertension (7.1%), lung diseases (7.1%), and hypercholesterolemia (6.3%). Gastrointestinal and liver diseases were the rarest, with prevalences of 3.1% and 1.3%, respectively.

The proportion of vascular diseases increased significantly from 42.6% in stage 1 to 68.2% in stages 3–4 ($p < 0.001$). Similarly, the proportion of obesity increased, from 25.7% in stage 1 to 84.5% in stages 3–4 ($p < 0.001$) (Table 3).

## Concomitant medications, conservative treatments, and surgical therapy

Table 4 summarises the most commonly reported concomitant medications and treatments from the medical history. Among conservative treatments, the most frequent was compression

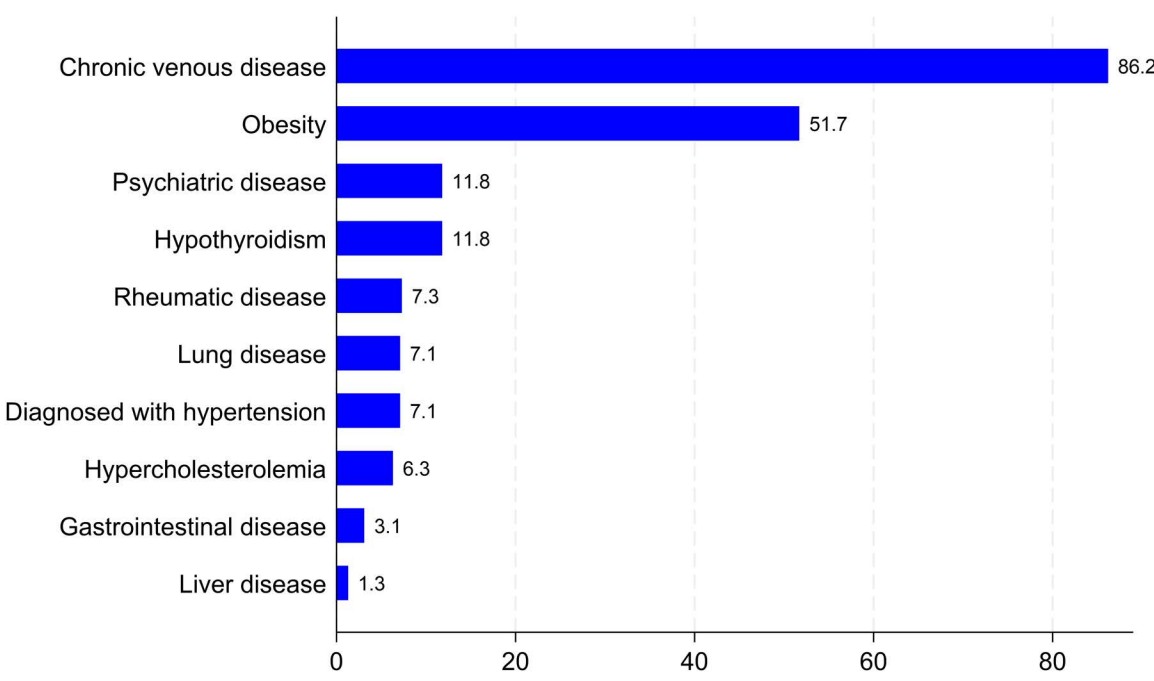

**Fig 3. Distribution of the most common types of comorbidities.**

**Table 3. Prevalence of comorbidities across different stages of lipedema.**

| Comorbidities | Stage 1 | Stage 2 | Stage 3-4 | p-value |
|---|---|---|---|---|
| | (N = 101) | (N = 169) | (N = 110) | |
| Vascular disease | 43 (42.6%) | 81 (47.9%) | 75 (68.2%) | <0.001 |
| Obesity | 26 (25.7%) | 78 (46.2%) | 93 (84.5%) | <0.001 |
| Hypercholesterolemia | 6 (5.9%) | 7 (4.1%) | 11 (10.0%) | 0.142 |
| Hypertension[a] | 3 (3.0%) | 9 (5.3%) | 15 (13.6%) | 0.005 |
| Lung disease | 7 (6.9%) | 12 (7.1%) | 8 (7.3%) | 0.995 |
| Gastrointestinal disease | 2 (2.0%) | 5 (3.0%) | 5 (4.5%) | 0.556 |
| Liver disease | 0 (0.0%) | 1 (0.6%) | 4 (3.6%) | 0.037 |
| Rheumatic disease | 5 (5.0%) | 9 (5.3%) | 14 (12.7%) | 0.038 |
| Psychiatric disease | 12 (11.9%) | 16 (9.5%) | 17 (15.5%) | 0.319 |
| Hypothyroidism | 5 (5.0%) | 19 (11.2%) | 21 (19.1%) | 0.006 |

[a]) The hypertension data in the table reflects the number of participants with confirmed hypertension diagnosis, rather than those identified through blood pressure measurements taken at baseline reported in Table 1.

therapy (55.9%), followed by manual lymphatic drainage (15.2%). Other commonly prescribed medications, beyond conservative treatments, included contraception (18.4), psychotropic drugs (12.6%), and hormonal therapy (11.0%).

Seventy-two patients (18.75%) underwent adjuvant liposuction. Mean volume of the lower limbs was 11396 mL before and 11099 mL after the intervention. For the upper limbs, mean volume was 3182 mL before and 2893 mL after the intervention (p < 0.0001) (Fig 4).

## Clinical characteristics associated with stages of lipedema

Table 5 summarizes the results of the logistic regression analysis. The univariate regression analysis revealed that age (OR: 1.07, 95% CI: 1.05-1.09, p < 0.000), BMI (OR: 1.24, 95% CI:

**Table 4. Treatment administered at baseline.**

| | Overall | Stage 1 | Stage 2 | Stage 3-4 | p-value |
|---|---|---|---|---|---|
| | (N = 381) | (N = 101) | (N = 169) | (N = 110) | |
| NSAID[a] | 26 (6.8%) | 5 (5.0%) | 12 (7.1%) | 9 (8.2%) | 0.639 |
| Psychotropic drugs | 48 (12.6%) | 9 (8.9%) | 24 (14.2%) | 15 (13.6%) | 0.418 |
| Anticoagulation | 11 (2.9%) | 1 (1.0%) | 2 (1.2%) | 8 (7.3%) | 0.005 |
| Antihypertensive | 23 (6.0%) | 1 (1.0%) | 8 (4.7%) | 14 (12.7%) | 0.001 |
| Analgesics | 21 (5.5%) | 3 (3.0%) | 10 (5.9%) | 8 (7.3%) | 0.376 |
| Contraception | 70 (18.4%) | 21 (20.8%) | 40 (23.7%) | 9 (8.2%) | 0.004 |
| Hormone therapy | 42 (11.0%) | 9 (8.9%) | 24 (14.2%) | 9 (8.2%) | 0.213 |
| Vasoactives | 21 (5.5%) | 6 (5.9%) | 12 (7.1%) | 3 (2.7%) | 0.288 |
| Lymphatic drainage | 58 (15.2%) | 15 (14.9%) | 26 (15.4%) | 16 (14.5%) | 0.981 |
| Compression | 213 (55.9%) | 66 (65.3%) | 83 (49.1%) | 63(57.3%) | 0.032 |

[a]) NSAID: Nonsteroidal anti-inflammatory drugs.

1.19-1.29, p < 0.000), and comorbidities (OR: 1.59, 95% CI: 1.39-1.81, p < 0.000) were significantly associated with the advanced stage of lipedema. In the multivariate analysis, only age (OR: 1.06, 95% CI: 1.04-1.08, p < 0.000) and BMI (OR: 1.22, 95% CI: 1.17-1.28, p < 0.000) remained significantly associated with advanced lipedema.

## Discussion

### Main findings

This study examined the clinical and epidemiological characteristics of 381 lipedema patients focusing on how their features relate to disease stage. Most patients reported disease onset during puberty, and about half had a family history of the condition. Nearly all experienced limb pain and disproportionate enlargement, with easy bruising and heaviness also common. Approximately half of the patients were classified as obese based on their BMI. Chronic venous disease affected the majority of patients, and thyroid disease and psychiatric disorders were commonly observed. Hypertension was relatively frequent, while diabetes, coronary artery disease, and cerebrovascular disease were exceptionally rare. Fatigue, anxiety, and depression were highly prevalent. Age, BMI, and the number of comorbidities tended to increase with disease stage, although multivariate analysis identified only age and BMI as significantly associated with advanced disease stage.

### Comparison with previous research

There is still much disagreement among experts in classifying lipedema and defining its typical characteristics [36]. Diagnosis still relies on clinical criteria originally proposed by Wold et al. in 1951 [26]. Following modifications introduced by Herbst [37], these criteria are still applied for routine diagnosis in several centers. American guidelines recommend classifying lipedema into five types and three stages (excluding stage 4) and suggest that symptoms are not essential for diagnosis. Certain comorbidities, such as joint hypermobility, are frequently reported and may support the diagnosis of lipedema. However, German guidelines emphasise that symptoms in affected areas are essential for diagnosis. Morphological staging is not a reliable indicator of disease severity [17], with classification better reflecting symptom severity and functional limitations. In our cohort, type III stage 2 lipedema was the most common presentation, consistent with findings from other studies [22–24]. Symptoms were also very common, particularly pain. We also found very high rates of chronic venous disease, as

## Limb Volume: Pre vs Post-Intervention

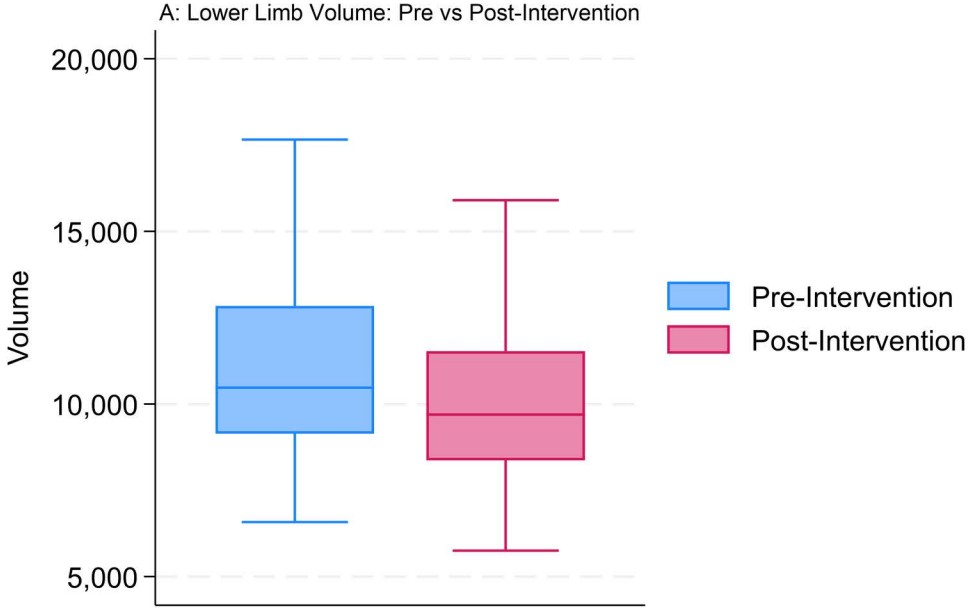

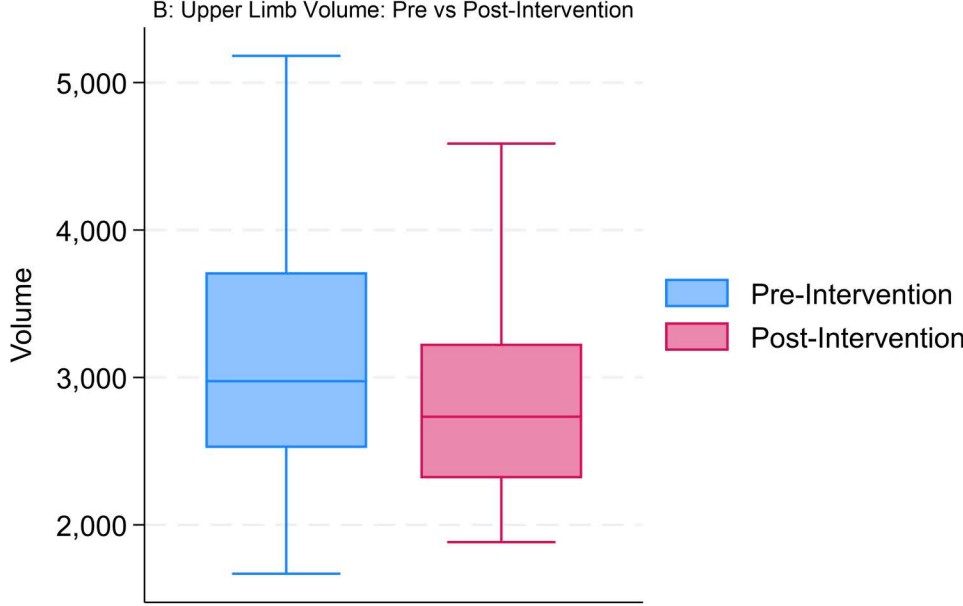

**Fig 4. Volume before and after surgical interventions: box plots with the median and interquartile range A: lower limbs; B: upper limbs.**

reported by others [22]. Interestingly, hypothyroidism was also one of the main comorbidities reported. This has also been described by others, though with higher rates, ranging from 20% [38] to > 30% [23,39,40]. Notably, joint hypermobility was uncommon in our cohort. A recent US study using a survey found that women with lipedema are more likely to present with joint hypermobility compared with healthy controls (OR, 12.88; 95% CI, 6.68–24.81; p < 0.0001)

**Table 5. Patient characteristics correlated with clinical stages of lipedema.**

| | Univariate logistic regression | | Multivariable logistic regression | |
|---|---|---|---|---|
| | OR (95% CI) | p - value | OR (95% CI) | p - value |
| Age | 1.07 (1.05 -1.09) | 0.000 | 1.06 (1.04 – 1.08) | 0.000 |
| BMI | 1.24 (1.19 - 1.29) | 0.000 | 1.22 (1.17 – 1.28) | 0.000 |
| Comorbidities | 1.59 (1.39 - 1.81) | 0.000 | 1.01 (0.86 - 1.19) | 0.834 |

[25], whereas other studies reported high rates of hypermobility in their cohorts of patients with lipedema[19,22,41,42]. Such conflicting results warrant further clarification.

The relationship between lipedema and obesity remains controversial, with obesity often reported as the most common associated condition. A UK study reported that 4% of lipedema patients had a normal weight, while 85% were obese [2]. A study conducted in the Netherlands reported that up to 80% of lipedema patients are overweight or obese [43]. Among 2,344 women at Lymphology clinics in 2015, 3% had a normal weight, 9% were overweight, and 88% were obese [44]. In our cohort, nearly 51% of patients had obesity class I or higher. BMI increased with disease stage, with obesity class III present in 31.5% of stage 3–4 patients compared to 1% in stage I. BMI was significantly correlated with advanced disease in multivariate analysis. Given that disease severity is typically linked to the amount of adipose tissue in affected limbs, these findings are not surprising and align with this well-established between disease severity and adipose tissue in affected limbs.

Defining obesity by BMI may be misleading in lipedema patients, as it includes pathological limb fat, leading to inflated values. The waist-to-height ratio (WHtR) offers a promising alternative, showing superior diagnostic accuracy for detecting obesity [45] and better predicting obesity-related outcomes in the general population [46]. In over 600 lipedema patients, Brenner et al. found that using WHtR instead of BMI reversed the obesity rates, reducing them from 51% to 17%. Some patients classified as overweight by BMI were reclassified as underweight by WHtR [47]. The latest German guidelines highlight the importance of WHtR for obesity detection, recommending the assessment of height and waist circumference at initial evaluation and follow-up in lipedema patients [17]. Waist measurements were unavailable for many patients in our cohort, limiting the use of WHtR. However, this metric is now routinely included in evaluations, and we aim to assess its effectiveness in identifying overweight and obesity without stigma in future analyses.

Interestingly, patients with lipedema may present a less severe cardiovascular profile [22,48]. Low prevalence of diabetes has been described in patients suffering from lipedema despite an average BMI of 39 ± 12 kg/m2 [42]. A study of 46 patients found most had a normal lipid profile, with only 11.7% having total cholesterol ≥ 240 mg/dL, compared to up to 33.5% in the general female population. Less than 30% of women with stage 2 or 3 lipedema had hypertension, which was absent in stage 1 patients. Data suggest hypertension rates of 32.4% in women aged 40–59 of any BMI, rising to 60% in Caucasian women with obesity and a mean age of [48]. In our study, cardiovascular risk factors and diseases were rare. This suggests that lipedema fat, typically gynoid in distribution, may offer protection against cardio-metabolic dysfunction.

Studies using validated questionnaires show that women with lipedema experience significantly greater psychological, emotional, and social impairments compared to the general population [38,49,50]. Furthermore, women with lipedema may suffer from depression, anxiety, and other forms of psychological disorders [38,39,49,51]. These were among the main comorbidities in our study. Notably, a dedicated questionnaire revealed depression rates more than twice those recorded in patient files, indicating many cases go undetected. The high

prevalence of depression and anxiety in our study population aligns with previous literature [23,39,40,52–54]. This is significantly higher than the estimated 14.4% prevalence (95% CI: 11.1%–11.7%) in the general female population [55]. Most studies link lipedema to psychological distress, though one study found it preceded pain occurrence in 80% of participants [54]. It remains unclear whether mental disorders are secondary to lipedema or have a different causal link. In our study, depression and anxiety were common comorbidities, highlighting the importance of assessing the psychological burden in lipedema patients and ensuring timely referral to specialist support when necessary.

In our cohort, age, BMI, and comorbidities were correlated with lipedema stage, though only age and BMI were significantly linked to advanced disease in multivariate analysis. The increase in comorbidities with disease stage likely reflects the patients' increasing age. The older age observed in individuals with advanced disease suggests a potential progression of lipedema, which contrasts with some previous suggestions. However, due to the design of our study, we cannot establish a causal relationship [56]). The latest German guidelines state that pain severity does not correlate with subcutaneous adipose tissue in the limbs[17]. In our cohort, pain, fatigue, anxiety, and depression levels did not differ significantly between stages, suggesting that current classifications fail to capture symptom severity and QoL impairment. Future clinical tools should include these parameters for improved classification.

## Strengths

Most clinical evidence on patients with lipedema comes from surveys or questionnaires [23–25,38–41,51,57], with fewer studies conducting detailed evaluations of cohorts of affected patients [22,42]. In contrast, our study makes a significant contribution by providing in-depth, systematic assessments of a well-defined cohort. This rigorous approach offers a more comprehensive and objective understanding of the clinical and epidemiological characteristics of lipedema, including its progression, severity, and associated factors. These strengths enhance the validity and applicability of our findings, addressing critical gaps in the current research.

## Limitations

Although retrospective studies are time-efficient and useful for studying rare diseases and outcomes, they are not without limitations [58]. The retrospective design may limit the generalisability of findings to broader populations, as data collection relies on previously recorded information, which may not account for all relevant variables [59]. Additionally, the single-centre setting could introduce biases related to clinical practices or patient demographics specific to our institution, potentially affecting the representativeness of the results [60]. Reliance on self-reported data for the questionnaires may further affect accuracy, as recall bias and subjectivity can influence responses [61]. Future studies should adopt multicentre designs and objective measures to improve validity and applicability.

## Conclusions

Our study highlights the clinical and psychological burden of lipedema, with widespread mental health issues such as anxiety, and depression. The correlation between disease stages, age, and BMI suggests potential disease progression over time, though this requires confirmation through prospective studies. The distribution of comorbidities underscores the need for tailored management. Pain, fatigue, anxiety, and depression levels showed no significant differences between stages, suggesting that current classifications may not adequately reflect symptom severity and QoL emphasising the need for clinical tools that integrate these parameters.

## Supporting information

**S1 Table. Symptoms across lipedema stages.**
(DOCX)

## Author contributions

**Conceptualization:** Xhyljeta Luta, Lucia Mazzolai.

**Data curation:** Xhyljeta Luta.

**Formal analysis:** Xhyljeta Luta.

**Investigation:** Xhyljeta Luta, Lucia Mazzolai.

**Methodology:** Xhyljeta Luta.

**Resources:** Lucia Mazzolai.

**Software:** Xhyljeta Luta.

**Supervision:** Lucia Mazzolai.

**Validation:** Xhyljeta Luta, Lucia Mazzolai.

**Visualization:** Xhyljeta Luta.

**Writing – original draft:** Xhyljeta Luta, Giacomo Buso, Lucia Mazzolai.

**Writing – review & editing:** Xhyljeta Luta, Giacomo Buso, Enrica Porceddu, Roxani Psychogyiou, Sanjiv Keller, Lucia Mazzolai.

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
