## [Decision Letter · Decision Letter 0]

29 Dec 2024

PONE-D-24-57104Clinical characteristics, comorbidities, and predictors of disease severity in lipedema: A retrospective study from a Swiss referral centrePLOS ONE

Dear Dr. Xhyljeta,

Thank you for submitting your manuscript to PLOS ONE. After careful consideration, we feel that it has merit but does not fully meet PLOS ONE’s publication criteria as it currently stands. Therefore, we invite you to submit a revised version of the manuscript that addresses the points raised during the review process.

Based on the reviewers' suggestions, the paper needs major revision.  The reviewers' comments can be found below.<o:p></o:p>

We look forward to receiving your revised manuscript.

Kind regards,

Tanja Grubić Kezele, Ph.D., M.D.

Academic Editor

PLOS ONE

Reviewers' comments:

Reviewer's Responses to Questions

**Comments to the Author**

1. Is the manuscript technically sound, and do the data support the conclusions?

Reviewer #1: Partly

Reviewer #2: Yes

2. Has the statistical analysis been performed appropriately and rigorously? 

Reviewer #1: Yes

Reviewer #2: Yes

3. Have the authors made all data underlying the findings in their manuscript fully available?

Reviewer #1: No

Reviewer #2: Yes

4. Is the manuscript presented in an intelligible fashion and written in standard English?

Reviewer #1: Yes

Reviewer #2: Yes

5. Review Comments to the Author

Reviewer #1: The big strength of the article is indeed the fact that it uses existing data of participants who have been diagnosed with lipedema.However, I have some comments that in my opinion may improve this article:

1. Title: you mention predictors of lipedema in the title, and later on in the text, and those predictors are age and BMI - I think taking into account the fact that this is retrospective and not longitudinal study it is really hard to say to what extent these are predictors, especially that BMI is such a problematic measurement. Maybe saying that these are correlates would just be safer and closer to the truth. 

Abstract:You mention:  "Descriptive statistics and logistic regression were used for analysis." Descriptive statistics in my opinion cannot be used to analyze but to describe - maybe you can rephrase this sentence?"Severe lipedema" - you use that term very often, I am not so sure if that's the best statement, maybe advanced lipedema? or some other term?I'd also rewrite the conclusions - I am not quite sure to what extent your study demonstrates clinical burden of lipedema and the sentence about overlooking symptoms is not clear to me - do you mean other studies or your study?  IntroductionAlthough there is not international consensus about management of lipedema - there are several guidelines in different countries, e.g. all of them mention the need for a complex approach, taking into account lifestyle management. Results,It is unclear to me what p-value means in Table one. Is the difference significant between all of the group, only one of them? What was the statistics used to calculate that, is it chi square? If so, then it should maybe be briefly explained in the text. Also, sometimes is reported, sometimes not, so it is just unclear why.Same in prevalence of comorbidities and differences between groups, was it calculated with chi-square?

  "The most common was compression therapy198 (55.9%), followed by contraception (18.4%), manual lymphatic drainage (15.2%), psychotropic199 medications (12.6%), and hormonal therapy (11.0%).  "Was contraception prescribed for lipedema? It is unclear to me why you report this, same with psychotropic drugs and hormonal therapy. Especially under the title conservative treatment. Maybe you need to clarify why you report this data and which of those treatments are for lipedema. 

Discussion

In the discussion you mention depression, however in the results you report that 64% reported anxiety, and 24 % depression, and you never refer to the anxiety in the discussion. It would be good to include this. 

You mention that  "Most studies link lipedema to psychological distress, though one study found it preceded lipedema symptoms in 80% of participants [52]"In the cited study the authors interviewed patients already diagnosed with ipedema and based on that concluded that stress preceded some symptoms (especially pain), I don't think we can say that stress or depression preceds lipedema symptoms based on such methodology. Or maybe we can say that is precedes pain occurrence. So I'd suggest to rephrase that. Based on your study you can safely say that there is a correlation, or that depression is frequent comorbidity. 

 " Moreover, the older age of those with severe disease supports the view that lipedema is progressive, contrary to some suggestions "

I think this might be a correlation - it is hard to say unless you conduct a longitudinal study. 

" In our cohort, pain, fatigue, anxiety, and depression levels did not differ significantly293 between stages, suggesting that current classifications fail to capture symptom severity and QoL"

I think this is an important conclusion that could be emphasized

Reviewer #2: the authors tackle a topic that is little researched, so the report is certainly of interest.

- I would use different words than "genetic transmission", being for sure a mutlifactorial disease

- In conservative treatment right nutrition must be considered, particularly ketogenic diet, that even in a case report, positively impacts QoL 10.3390/life11121402

- criteria (2) is a poor indicator, in clinical practice there are good results reported if right nutritional plan are applied, not the case of "classical" scheme with 50% of calories from carbohydrate or more

- two comorbidities are often reported: PCOS and insulin resistance, have you been screened?

6. PLOS authors have the option to publish the peer review history of their article (what does this mean? ). If published, this will include your full peer review and any attached files.

**Do you want your identity to be public for this peer review?** For information about this choice, including consent withdrawal, please see our Privacy Policy .

Reviewer #1: No

Reviewer #2: No

---

## [Author Response · Author response to Decision Letter 0]

16 Jan 2025

We thank the editor and the reviewers for their time and thoughtful comments that have helped us to improve our manuscript. Our detailed responses below are bolded directly underneath each Editor/Reviewer comment. Corresponding changes are made in the main body of the manuscript.

We have ensured that our revised manuscript adheres to PLOS ONE's style requirements. We have referred to the style templates provided at the suggested links.

Our study involves sensitive human participant data, and ethical restrictions prevent us from sharing de-identified datasets publicly. These restrictions were imposed by Research Ethics Committee. Requests for data access can be directed to the following contact referencing our project ID: CER-VD, BASEC 2023-02221:

Commission cantonale d'éthique de la recherche sur l'être humain

Avenue de Chailly 23

1012 Lausanne

Tel: + 41 21 316 18 36

E-Mail: scientifique.cer@vd.ch

We have updated our Data Availability Statement in the submission form to reflect this restriction and provide clear instructions for requesting data access.

Reviewers' comments:

Authors’ response to Reviewer #1

The big strength of the article is indeed the fact that it uses existing data of participants who have been diagnosed with lipedema. However, I have some comments that in my opinion may improve this article:

We thank the reviewer for the detailed and valuable suggestions.

Title: you mention predictors of lipedema in the title, and later on in the text, and those predictors are age and BMI - I think taking into account the fact that this is retrospective and not longitudinal study it is really hard to say to what extent these are predictors, especially that BMI is such a problematic measurement. Maybe saying that these are correlates would just be safer and closer to the truth.

We agree with the reviewer and have revised the title to replace "predictors" with "correlates" to better reflect the nature of our analysis.

Clinical characteristics, comorbidities, and correlation with advanced lipedema stages: A retrospective study from a Swiss referral centre

Abstract: You mention: "Descriptive statistics and logistic regression were used for analysis." Descriptive statistics in my opinion cannot be used to analyze but to describe - maybe you can rephrase this sentence?

We have revised the sentence in the abstract to clarify that descriptive statistics were used to summarise the data, while logistic regression was employed for analysis.

Descriptive statistics were used to summarise the data, and logistic regression was employed for analysis (Page 2, lines 31-32).

"Severe lipedema" - you use that term very often, I am not so sure if that's the best statement, maybe advanced lipedema? or some other term?

We agree with the suggestion and have replaced "severe lipedema" with "advanced lipedema" throughout the manuscript.

I'd also rewrite the conclusions - I am not quite sure to what extent your study demonstrates clinical burden of lipedema and the sentence about overlooking symptoms is not clear to me - do you mean other studies or your study?

We have rephrased the conclusions in the abstract to represent our findings, emphasizing the correlation more accurately between advanced disease stages and quality of life and clarifying the reference to overlooked symptoms .

Our study highlights frequent comorbidities in patients with lipedema, including chronic venous disease, obesity, and mental health conditions such as anxiety and depression. The distribution of comorbidities supports the need for tailored management. The correlation between disease stages, age, and BMI suggests potential progression, warranting confirmation through prospective studies (Page3, lines 46-49).

Introduction: Although there is not international consensus about management of lipedema - there are several guidelines in different countries, e.g. all of them mention the need for a complex approach, taking into account lifestyle management.

We have expanded the introduction to reference international guidelines on lipedema management and emphasized the importance of a multifaceted approach.

Although there is no international consensus on the management of lipedema, several guidelines exist across different countries. These guidelines universally emphasise the importance of a comprehensive approach that includes lifestyle management (Page 4, lines 71-74).

References

• Reich-Schupke S, Schmeller W, Brauer WJ, Cornely ME, Faerber G, Ludwig M, Lulay G, Miller A, Rapprich S, Richter DF, Schacht V, Schrader K, Stücker M, Ure C. S1 guidelines: Lipedema. J Dtsch Dermatol Ges. 2017 Jul;15(7):758-767.

• Faerber G, Cornely M, Daubert C, et al. S2k guideline lipedema. JDDG: Journal der Deutschen Dermatologischen Gesellschaft. 2024; 22: 1303–1315.

• Hardy D, Williams A. Best practice guidelines for the management of lipoedema. Br J Community Nurs. 2017 Oct 1;22(Sup10):S44-S48. doi: 10.12968/bjcn.2017.22.Sup10.S44. PMID: 28961048.

• Herbst KL, Kahn LA, Iker E, Ehrlich C, Wright T, McHutchison L, Schwartz J, Sleigh M, Donahue PM, Lisson KH, Faris T, Miller J, Lontok E, Schwartz MS, Dean SM, Bartholomew JR, Armour P, Correa-Perez M, Pennings N, Wallace EL, Larson E. Standard of care for lipedema in the United States. Phlebology. 2021 Dec;36(10):779-796. doi: 10.1177/02683555211015887. Epub 2021 May 28. PMID: 34049453; PMCID: PMC8652358.

• Cannataro R, Michelini S, Ricolfi L, Caroleo MC, Gallelli L, De Sarro G, Onorato A, Cione E. Management of Lipedema with Ketogenic Diet: 22-Month Follow-Up. Life (Basel). 2021 Dec 15;11(12):1402. doi: 10.3390/life11121402. PMID: 34947933; PMCID: PMC8707844.

Results: It is unclear to me what p-value means in Table one. Is the difference significant between all of the group, only one of them?

We apologise for any confusion. The p-value in Table 1 represents the differences between individual groups rather than an overall comparison. To make this clearer, we have added a note in Table 1 explaining the interpretation of the p-values provided (Page 9, lines 160-161).

What was the statistics used to calculate that, is it chi square? If so, then it should maybe be briefly explained in the text. Also, sometimes is reported, sometimes not, so it is just unclear why. Same in prevalence of comorbidities and differences between groups, was it calculated with chi-square?

The statistics used to calculate the p-values in Table 1 were indeed derived from chi-square tests. We have added a brief explanation of the statistical method in the text to provide clarity.

Comparisons across lipedema stages were conducted using chi-square tests for categorical variables (e.g., BMI classification, lipedema type, and comorbidities) (Page 7, lines 138-140).

"The most common was compression therapy198 (55.9%), followed by contraception (18.4%), manual lymphatic drainage (15.2%), psychotropic199 medications (12.6%), and hormonal therapy (11.0%). "Was contraception prescribed for lipedema? It is unclear to me why you report this, same with psychotropic drugs and hormonal therapy. Especially under the title conservative treatment. Maybe you need to clarify why you report this data and which of those treatments are for lipedema.

We acknowledge the confusion and have clarified that these treatments were part of the participants' reported medical history and not specifically prescribed for lipedema. Additionally, we have updated the heading title to: Concomitant medications, conservative treatments, and surgical therapy (Page 11, line 203).

Table 4 summarises the most commonly reported concomitant medications and treatments from the medical history. Among conservative treatments, the most frequent was compression therapy (55.9%), followed by manual lymphatic drainage (15.2%). Other commonly prescribed medications, beyond conservative treatments, included contraception (18.4), psychotropic drugs (12.6%), and hormonal therapy (11.0%) (Page 11, lines 204-208).

Discussion

In the discussion you mention depression, however in the results you report that 64% reported anxiety, and 24 % depression, and you never refer to the anxiety in the discussion. It would be good to include this.

Thank you for your observation. We have expanded the discussion to incorporate the data on anxiety and highlighted its significance in relation to previous research.

The high prevalence of depression and anxiety in our study population aligns with previous literature (Page 15, lines 289-290).

You mention that "Most studies link lipedema to psychological distress, though one study found it preceded lipedema symptoms in 80% of participants [52]"In the cited study the authors interviewed patients already diagnosed with lipedema and based on that concluded that stress preceded some symptoms (especially pain), I don't think we can say that stress or depression precedes lipedema symptoms based on such methodology. Or maybe we can say that is precedes pain occurrence. So I'd suggest to rephrase that.

Thank you for bringing this to our attention. We agree with your observation and have revised the statement to clarify that the cited study suggests stress may precede the occurrence of certain symptoms, particularly pain, rather than lipedema symptoms more broadly.

Most studies link lipedema to psychological distress, though one study found it preceded pain occurrence in 80% of participants (Page 15, lines 291-293).

Based on your study you can safely say that there is a correlation, or that depression is frequent comorbidity.

We agree that our study demonstrates a correlation and confirms that depression and anxiety are frequent comorbidities. We have updated the text to reflect this and further emphasised the importance of addressing the psychological burden in lipedema patients through timely assessment and specialist support.

In our study, depression and anxiety were common comorbidities, highlighting the importance of assessing the psychological burden in lipedema patients and ensuring timely referral to specialist support when necessary (Page 15, lines 294-296).

" Moreover, the older age of those with severe disease supports the view that lipedema is progressive, contrary to some suggestions "

I think this might be a correlation - it is hard to say unless you conduct a longitudinal study.

We have revised the discussion to clarify that our findings suggest a correlation between age and advanced disease stage, without implying definitive evidence of disease progression.

The older age observed in individuals with advanced disease suggests a potential progression of lipedema, which contrasts with some previous suggestions. However, due to the design of our study, we cannot establish a causal relationship (Pages 15-16, lines 299-301).

" In our cohort, pain, fatigue, anxiety, and depression levels did not differ significantly293 between stages, suggesting that current classifications fail to capture symptom severity and QoL"

I think this is an important conclusion that could be emphasized.

We have highlighted this finding more prominently in conclusions, stressing the limitations of current classification systems in capturing symptom severity and quality of life.

Pain, fatigue, anxiety, and depression levels showed no significant differences between stages, suggesting that current classifications may not adequately reflect symptom severity and QoL emphasising the need for clinical tools that integrate these parameters (Page 17, lines 328-331).

Authors’ response to Reviewer #2

Reviewer #2: the authors tackle a topic that is little researched, so the report is certainly of interest.

We thank the reviewer for the supportive comments.

- I would use different words than "genetic transmission", being for sure a mutlifactorial disease .

We have replaced "genetic transmission" with "genetic predisposition" to better align with the multifactorial nature of lipedema.

Genetic predisposition is possible, as lipedema often runs in families (Page 3, lines 57-58).

- In conservative treatment right nutrition must be considered, particularly ketogenic diet, that even in a case report, positively impacts QoL 10.3390/life11121402

We acknowledge the potential role of nutrition, particularly the ketogenic diet, in improving the quality of life for lipedema patients, as highlighted in the cited case report. While our study did not specifically explore dietary interventions, we agree that this is an important area for further research to better understand its impact on symptom management and overall quality of life. To highlight the significance of nutrition in conservative treatment approaches, we have incorporated the following references into the introduction:

• Cannataro R, Michelini S, Ricolfi L, Caroleo MC, Gallelli L, De Sarro G, et al. Management of Lipedema with Ketogenic Diet: 22-Month Follow-Up. Life (Basel). 2021;11(12).

• Lundanes J, Sandnes F, Gjeilo KH, Hansson P, Salater S, Martins C, Nymo S. Effect of a Low-Carbohydrate Diet on Pain and Quality of Life in Female Patients with Lipedema: A Randomized Controlled Trial. Obesity (Silver Spring). 2024 Jun;32(6):1071-1082. doi: 10.1002/oby.24026. Epub 2024 Apr 16. PMID: 38627016.

•

We appreciate the perspective on the benefits of tailored nutritional plans over traditional dietary approaches. In addition, we have revised Criterion 2 to reflect these considerations and to emphasize the potential effectiveness of specific nutritional interventions:

(2) Minimal improvement with conventional weight loss strategies (excluding ketogenic diets) (Page 5, lines 94-95).

- two comorbidities are often reported: PCOS and insulin resistance, have you been screened?

We acknowledge the importance of screening for PCOS and insulin resistance in the context of lipedema. However, these comorbidities were not screened for in our study. We appreciate your suggestion and will consider including such assessments in future research to provide a more comprehensive analysis.

---

## [Decision Letter · Decision Letter 1]

28 Jan 2025

Clinical characteristics, comorbidities, and correlation with advanced lipedema stages: A retrospective study from a Swiss referral centre

PONE-D-24-57104R1

Dear Dr. Xhyljeta,

We’re pleased to inform you that your manuscript has been judged scientifically suitable for publication and will be formally accepted for publication once it meets all outstanding technical requirements.

Kind regards,

Tanja Grubić Kezele, Ph.D., M.D.

Academic Editor

PLOS ONE

Additional Editor Comments (optional):

Reviewers' comments:

Reviewer's Responses to Questions

**Comments to the Author**

1. If the authors have adequately addressed your comments raised in a previous round of review and you feel that this manuscript is now acceptable for publication, you may indicate that here to bypass the “Comments to the Author” section, enter your conflict of interest statement in the “Confidential to Editor” section, and submit your "Accept" recommendation.

Reviewer #1: All comments have been addressed

Reviewer #2: All comments have been addressed

2. Is the manuscript technically sound, and do the data support the conclusions?

Reviewer #1: Yes

Reviewer #2: Yes

3. Has the statistical analysis been performed appropriately and rigorously? 

Reviewer #1: Yes

Reviewer #2: Yes

4. Have the authors made all data underlying the findings in their manuscript fully available?

Reviewer #1: Yes

Reviewer #2: Yes

5. Is the manuscript presented in an intelligible fashion and written in standard English?

Reviewer #1: Yes

Reviewer #2: Yes

6. Review Comments to the Author

Reviewer #1: (No Response)

Reviewer #2: The authors addressed engough my comments

7. PLOS authors have the option to publish the peer review history of their article (what does this mean? ). If published, this will include your full peer review and any attached files.

**Do you want your identity to be public for this peer review?** For information about this choice, including consent withdrawal, please see our Privacy Policy .

Reviewer #1: No

Reviewer #2: No

---

## [Editor Report · Acceptance letter]

PONE-D-24-57104R1

PLOS ONE

Dear Dr. Luta,

I'm pleased to inform you that your manuscript has been deemed suitable for publication in PLOS ONE. Congratulations! Your manuscript is now being handed over to our production team.

Kind regards,

on behalf of

Prof. dr. Tanja Grubić Kezele

Academic Editor

PLOS ONE